# Fabrication and Characterization of Polysaccharide Metallohydrogel Obtained from Succinoglycan and Trivalent Chromium

**DOI:** 10.3390/polym13020202

**Published:** 2021-01-08

**Authors:** Dajung Kim, Seonmok Kim, Seunho Jung

**Affiliations:** 1Department of Bioscience and Biotechnology, Microbial Carbohydrate Resource Bank (MCRB), Konkuk University, Seoul 05029, Korea; dajung903@naver.com (D.K.); gkdurk9999@naver.com (S.K.); 2Center for Biotechnology Research in UBITA (CBRU), Institute for Ubiquitous Information Technology and Applications (UBITA), Department of Systems Biotechnology, Konkuk University, Seoul 05029, Korea

**Keywords:** hydrogels, succinoglycan, trivalent chromium, metallohydrogel, rheological properties

## Abstract

In the present study, a polysaccharide metallohydrogel was successfully fabricated using succinoglycan and trivalent chromium and was verified via Fourier transform infrared spectroscopy, differential scanning calorimetry analysis, thermogravimetric analysis (TGA), field emission scanning electron microscopy, and rheological measurements. Thermal behavior analysis via TGA indicated that the final mass loss of pure succinoglycan was 87.8% although it was reduced to 65.8% by forming a hydrogel with trivalent chromium cations. Moreover, succinoglycan-based metallohydrogels exhibited improved mechanical properties based on the added concentration of Cr^3+^ and displayed a 10 times higher compressive stress and enhanced storage modulus (G′) of 230% at the same strain. In addition, the pore size of the obtained SCx could be adjusted by changing the concentration of Cr^3+^. Gelation can also be adjusted based on the initial pH of the metallohydrogel formulation. This was attributed to crosslinking between chromium trivalent ions and hydroxyl/carboxyl groups of succinoglycan, each of which exhibits a specific pH-dependent behavior in aqueous solutions. It could be used as a soft sensor to detect Cr^3+^ in certain biological systems, or as a soft matrix for bioseparation that allows control of pore size and mechanical strength by tuning the Cr^3+^ concentration.

## 1. Introduction

Microbial extracellular polysaccharides (EPS) are biodegradable and biocompatible polymers with the potential to replace petroleum-derived substances, such as acrylic fiber and polyurethane. Their ability to transform into porous-forming gels is attracting significant attention [1,2,3]. Polysaccharide-based gels are used in various applications including catalyst support, drug delivery agents, and water production control [4,5,6,7]. Additionally, EPS has scientific and commercial requirements owing to its wide range of industrial applications and biological properties. They are extensively used in the food industry as gelling and thickening agents, emulsifiers, and texture modifiers to replace commercial hydrocolloids [8,9]. Various applications of microbial EPS, such as chitosan, alginate, galactan, pullulan, curdlan, xanthan, and gellan are widely reported [10,11,12,13,14,15]. Succinoglycan is an acidic extracellular polysaccharide of octasaccharide that consists of a 1:7 ratio of galactose and glucose and is produced by soil bacteria such as *Sinorhizobium* and *Agrobacterium.* It also contains non-carbohydrate substituents including succinate, pyruvate, and acetate groups [16,17,18]. It is resistant to high temperatures and high salt or ionic concentrations and has high viscosity [19]. This property of succinoglycan holds immense potential in commercial applications such as cosmetics and household products, fertilizer formulations, pharmaceuticals, emulsifiers, gelling agents, stabilizers, and thickeners [20].

An approach to extend the functionality of polysaccharide hydrogels involves using dynamic non-covalent interactions via crosslinking through metal ions [21], and several studies examined gelation via the co-ordination of various acidic polysaccharides and metal ions [22,23,24,25,26,27]. Diverse multivalent metal ions, such as Fe^3+^, Al^3+^, Ca^2+^, Cu^2+^, Mn^2+^, Co^2+^, and Cr^3+^ (which can be co-ordinated with the functional groups of acidic polysaccharides), have been reported [22]. For example, alginate is the most widely examined acidic polysaccharide and creates a specific gelling structure, termed as the “egg-box” model, by the interaction of Ca^2+^ and carboxylic groups of guluronate blocks [28]. In addition to Ca^2+^, transition metal ions, such as Fe^3+^, were also observed to form metallohydrogels with alginates [29,30], where metal co-ordinations provide new mechanical properties by controlling the metal co-ordination conditions [31]. Carboxymethyl chitosan-metal ions (e.g., Ag^+^, Cu^2+^, and Zn^2+^) hydrogels exhibit excellent antibacterial activity against *S. aureus* and *E. coli* and display enhanced mechanical behavior [32]. Other microbial acidic polysaccharides, gellan [33], salecan [34], and xanthan [35,36] also show that divalent or trivalent cations can induce gelation via chemical crosslinking between carboxylic (–COO^−^) groups and metal ions. In particular, xanthan and salecan enhance metallo-gelation by adding trivalent chromium cations [34,35,36]. Moreover, Cr^3+^ is widely considered to exhibit significantly lower toxicity than Cr^6+^ [37,38,39]. Therefore, several application studies involving living cells using Cr^3+^ are also reported [40,41,42].

A salecan-based hydrogel co-ordinated with Cr^3+^ as a crosslinking agent affects the expansion capacity, pore size, and mechanical strength of the hydrogel by changing the Cr^3+^ concentration [34]. In the study of xanthan/Cr^3+^ systems, the gelation time changed based on the change in pH [43], where the properties of these crosslinks, formed primarily through the formation of chemical bonds between the Cr^3+^ and acidic carboxylic (–COO) groups of xanthan, were relatively strong [44]. Therefore, this pH-controllable gelation in xanthan-based metallohydrogels suggests a potential application to selectively block water during oil recovery operations [45]. Given that succinoglycan has many carboxyl groups as non-carbohydrate substituents that can be potential functional groups via metal co-ordination, it also exhibits the potential to serve as succinoglycan-based metallohydrogels with metal ions.

In this paper, we report the characteristic properties of succinoglycan metallohydrogels (SCx) induced by trivalent chromium (Cr^3+^). Various investigations, such as rheometry, Fourier transform infrared (FTIR) spectroscopy, X-ray diffraction measurements (XRD), thermogravimetric analysis (TGA), differential scanning calorimetry (DSC), and field emission scanning electron microscopy (FESEM), were performed to characterize SCx.

## 2. Materials and Methods

### 2.1. Materials

*Sinorhizobium meliloti* Rm1021 was supplied by the Microbial Carbohydrate Resource Bank at Konkuk University, Korea. Chromium(III) chloride hexahydrate (95%) was purchased from Daejung Chemicals & Metals Co., Ltd. (Siheung, Korea). All other chemicals were of analytical grade and used without further purification. To vary the pH of the samples, sodium hydroxide (Daejung Chemicals & Metals Co., Ltd., Siheung, Korea, 97%) and hydrochloric acid, as a titration standard for preparation of 1 M or 0.1 M solution (Daejung Chemicals & Metals Co., Ltd., Siheung, Korea, 35–37%), were used.

### 2.2. Growth Conditions and Production of Succinoglycan

Succinoglycan was isolated and purified from *S. meliloti* Rm1021, as previously described. Bacteria were cultured in glutamic acid-mannitol-salts medium, which was adjusted to a pH of 7.00 at 30 °C for 7 days with shaking (180 rpm). After 7 days, the cells were centrifuged at 8000× *g* for 15 min at 4 °C, and the supernatant was collected. Three volumes of ethanol were added to the supernatant to obtain succinoglycan. Furthermore, succinoglycan was first precipitated, and the precipitate was subsequently dissolved in distilled water and then dialyzed (MWCO 12–14 kDa, distilled water for 3 days). After collection, the purified succinoglycan was freeze-dried for later use. The molecular weights of succinoglycan were estimated via gel permeation chromatography (GPC) analysis performed at 30 °C. GPC was performed using a Waters Breeze System equipped with a Waters 1525 Binary pump and a Waters 2414 refractive index detector using 0.02 N sodium nitrate as a solvent at a flow rate of 0.8 mL min^−1^. The molecular weight (Mw) of succinoglycan—as estimated via GPC—is 1.8 × 10^5^ Da, as listed in Appendix A.

### 2.3. Preparation for Gelation Confirmation Experiment

To examine the effect of various metal ions, stock aqueous solutions of succinoglycan (aq, 2%) and 0.25 M of KCl, NaCl, CaCl_2_, MgCl_2_, CuCl_2_, ZnCl_2_, AlCl_3_, FeCl_3_, and CrCl_3_ were prepared. The metal ionic solution was added to the prepared succinoglycan solution and stirred for 3 min. The resulting mixture was poured into a mold and left for 24 h.

### 2.4. Preparation of Hydrogels

Succinoglycan solution (aq, 2%) was mixed with an equal volume of Cr^3+^ aqueous solution. Additionally, Cr^3+^ aqueous solution (final 6.6–52.8 mM) was prepared by dispersing a defined amount of CrCl_3_ 6H_2_O in deionized water under moderate stirring. It was then left for 1 day to realize hydrolysis. The final stock solution exhibited a purple–green color. The mixture was stirred at room temperature for 3 min and then left for at least 18 h to spontaneously form succinoglycan/Cr^3+^ co-ordination hydrogels. The resulting metallohydrogel was formed by mixing an aqueous succinoglycan solution with a Cr^3+^ aqueous solution. It was labeled as SCx, where x corresponded to 6.6, 13.2, 26.4, and 52.8 denoting 6.6, 13.2, 26.4, and 52.8 mM of Cr^3+^ molar concentrations, respectively. A small amount of acid or alkali (0.1 M HCl or 0.1 M NaOH) was added to the gel initially obtained at pH 5 to determine the effect of pH on gelation. Moreover, the effect of pH on uncrosslinked succinoglycan solutions was investigated by adding HCl and NaOH to a mixture of succinoglycan and Cr^3+^ initially obtained at pH 1 and pH 9.

### 2.5. Fourier Transform Infrared (FTIR) Spectroscopy

The FTIR spectra of the lyophilized succinoglycan and SCx were recorded via an FTIR spectrometer (Spectrum Two FT-IR, Perkin Elmer, Waltham, MA, USA) with a PIKE MIRacle ATR accessory at a resolution of 1 cm^−1^ using eight scans in the wavenumber range of 4000–600 cm^−1^.

### 2.6. X-ray Diffraction Measurements (XRD)

The XRD pattern was measured via a Rigaku SmartLab 9 kW X-ray diffractometer, and a HyPix-3000 was used as the detector for analysis. Samples were used after lyophilization. Tests were performed at 30 kV and 20 mA, and scans were performed in the 2θ range of 10–80 °C.

### 2.7. Thermogravimetric Analysis (TGA)

Thermogravimetric analysis (TGA) was performed using a Perkin Elmer Pyris1 thermogravimetric analyzer. The thermal analyzer was operated under nitrogen atmospheric pressure and regulated by compatible PC commands. The dried sample (10 mg) was placed in a crucible and heated in an enclosed system with a linear temperature increase at a rate of 10 °C/min over a temperature range of 25 to 600 °C.

### 2.8. Differential Scanning Calorimetry (DSC)

DSC was performed via a Discovery DSC apparatus (TA Instruments, New Castle, DE, USA). The DSC samples (10 mg) were weighed on an aluminum pan and the pan was sealed. The sample was scanned under a purified nitrogen atmosphere at a heating rate of 10 °C/min from 25 to 180 °C.

### 2.9. Field Emission Scanning Electron Microscopy (FESEM)

The cross-sectional morphologies of pure succinoglycan and SCx were observed via FESEM (JSM-7800F Prime, JEOL Ltd., Akishima, Japan). The samples were rapidly frozen and lyophilized for 24 h. For observation, the surface of the cross-sectioned hydrogel was coated with a thin layer of platinum at 10 mA for 60 s in vacuum to render them electrically conductive.

### 2.10. Rheological Experiments

The rheological properties of the hydrogels were analyzed via oscillation angular frequency sweep and temperature ramp tests of succinoglycan solution that were performed using a DHR-2 rheometer (TA Instruments, New Castle, DE, USA) equipped with 20 mm parallel plates. The angular frequency was swept from 0.1 to 100 rad/s at a strain of 0.5% at a constant angular frequency of 10 rad/s and a constant strain of 1.0%. A stress–strain amplitude sweep test was conducted on samples from 0.1% to a maximum strain of 100% at 1.0 Hz to determine the limit of the linear viscoelastic region. Each measurement was performed in triplicate.

### 2.11. Compressive Test

With respect to the compressive test, hydrogel discs with a height of 10 mm and a diameter of 20 mm were prepared. Furthermore, compressive tests were performed using an Instron E3000LT (Instron Inc., Norwood, MA, USA). The sample was placed on a plate and compressed at a rate of 5 mm/min. Compressive stress was recorded when the sample was compressed with a strain of 40%. The measurement was performed in triplicate.

### 2.12. Equilibrium Swelling Ratio

The dried metallohydrogel samples were precisely weighed and immersed in deionized water at 25 °C. At each predetermined time interval, the SCx was removed from the swelling medium and weighed after removing excess surface water. The measurement was continued until constant weights of the swollen metallohydrogels were obtained. The water swelling ratios were calculated as follows [46]:(1)$$Swelling\;ratio=\;\frac{W_{s}-W_{d}}{W_{d}}$$
where *W_s_* denotes the weight of the swollen hydrogel, and *W_d_* denotes the weight of the dried hydrogel. Each measurement was performed in triplicate.

### 2.13. UV–Vis Spectrophotometer

A UV–Vis spectrophotometer (UV 2450, Shimadzu Corporation, Kyoto, Japan) was used to confirm that chromium ions were released after immersing the SCx in distilled water (D.W), and spectra were observed from 200 to 700 nm at 2-h intervals over 48 h.

## 3. Results

### 3.1. Fabrication of Metallohydrogels

Various aqueous metal ionic solutions, including K^+^, Na^+^, Ca^2+^, Mg^2+^, Cu^2+^, Zn^2+^, Al^3+^, Fe^3+^, and Cr^3+^, were prepared to determine metal ions capable of forming hydrogels mixed with an aqueous succinoglycan (Figure 1) solution. Figure 2a shows the results of the inverted vial tests for each mixed solution corresponding to the different metal ions. Most metal ions, with the exception of Fe^3+^ and Cr^3+^, did not induce physical gels. However, stronger physical gel formation was observed only when the Cr^3+^ solution was added to the succinoglycan aqueous solution when compared to the Fe^3+^ solution. Furthermore, this was confirmed by measuring the difference between the storage modulus (G′) and loss modulus (G″) via rheological experiments (Figure 2b). The result showed that succinoglycan induce effective metallohydrogel formation by adding Cr^3+^ cations. As listed in Table 1, a succinoglycan-based metallohydrogel is prepared and labeled as SCx, where x corresponds to the aqueous Cr^3+^ molar concentration. 

### 3.2. Characterization of Metallohydrogels

The preparation of metallohydrogel was confirmed via FTIR analysis [47]. The metallic co-ordination of succinoglycan with chromium trivalent cations was confirmed based on the differences in the characteristic absorption peaks of succinoglycan and succinoglycan-based metallohydrogel (SCx). The explanation for the adjustment was typically set to SC_26.4_, and the same peak shift trend occurred for the other SCx. As shown in Figure 3, the FTIR spectrum of succinoglycan exhibited an absorption peak at 3284 cm^−1^, which corresponded to the O–H stretching band, while the C=O stretching carbonyl ester of the acetate group exhibited an absorption peak at 1728 cm^−1^. The absorption peaks at 1626 and 1382 cm^−1^ were attributed to the asymmetric C=O stretching vibration of the succinate and pyruvate functional groups and symmetric stretching vibration of the carboxylate –COO– group from acid residues, respectively [48]. Additionally, the absorption peak at 1082 cm^−1^ was attributed to the asymmetrical C–O–C stretching vibration [47,48,49]. However, when succinoglycan was mixed with Cr^3+^ cations, SC_26.4_ absorption peaks at 3284, 1626, and 1082 cm^−1^ were shifted to 3317, 1634, and 1050 cm^−1^, respectively. This was potentially due to the co-ordination of succinoglycan with Cr^3+^ ions. The shifts in the absorption peaks indicated that interactions occurred between the carboxyl groups of succinoglycan and Cr^3+^ cations. When succinoglycan was co-ordinated with Cr^3+^ cations, the –OH stretching band at 3284 cm^−1^ of succinoglycan was red-shifted significantly to 3317 cm^−1^ due to the metal co-ordination induced by Cr^3+^. Furthermore, the C=O stretching vibration peak at 1626 cm^−1^ (which was related to succinate and pyruvate within the succinoglycan) was shifted to 1634 cm^−1^. Additionally, the asymmetrical C–O–C stretching peak by the backbone sugar shifted from 1082 to 1050 cm^−1^. The shifts of the peaks at 1626 and 1082 cm^−1^ were attributed to the co-ordination of the carboxyl group with Cr^3+^ cations. Moreover, the results also indicated that the –OH group of succinoglycan was mainly involved in co-ordination with the Cr^3+^ cation via the larger shift of the –OH stretching band of succinoglycan. In conclusion, the results indicated that Cr^3+^ cations are strongly co-ordinated with the carboxyl and hydroxyl groups of succinoglycan.

Figure 4 shows a schematic of the metallohydrogel formation induced by the co-ordination of Cr^3+^ with succinoglycan.

### 3.3. Analysis of Crystallinity and Diffraction Angle of Metallohydrogels

X-ray diffraction (XRD) measurements were obtained to examine the molecular crystallinity of SCx. As shown in Figure 5, succinoglycan exhibited a large selection at 2θ = 20.5°. The diffraction peak of SCx was similar to that of succinoglycan. However, increases in the strength of SCx from SC_6.6_ to SC_52.8_ decreased crystallinity. The amorphous region of the succinoglycan polymer matrix was augmented by the addition of Cr^3+^. This indicates that succinoglycan changes from a crystalline to an amorphous state, thereby indicating that intermolecular hydrogen bonds between metal cations and hydroxyl-/carboxyl groups of succinoglycan occurred for a metallohydrogel [50,51,52]. The results indicated that the addition of Cr^3+^ changes the intrinsic crystallinity of succinoglycan and generates another regular molecular arrangement.

### 3.4. Thermal Analysis of Metallohydrogels

#### 3.4.1. Thermogravimetric Analysis (TGA)

Figure 6 shows TGA and differential thermogravimetry (DTG) diagrams of succinoglycan and SCx. As shown in Figure 6a, the thermal degradation of succinoglycan is divided into two stages. The first mass loss explained the loss of hydrogen bonds and absorbed water molecules attached to the carboxyl group of succinoglycan. The initial moisture content of the samples was attributed to high levels of carboxyl groups of succinoglycan interacting with water molecules [53]. The result explains the high moisture and water affinity of succinoglycan given its high content of carboxyl groups [54]. In the first stage, the mass loss of succinoglycan and SCx was 8.05% below 95 °C and 10–20% at approximately 110 °C, respectively (Table 2). In the second stage, succinoglycan exhibited a mass loss of 60.64% in the temperature range of 246.14–370.80 °C. In the case of SCx, a second-stage temperature was formed at a lower temperature than that of succinoglycan. With respect to SC_13.2_, a loss of 53.08% occurred in the temperature range of 213.12–477.24 °C. In particular, SC_26.4_ and SC_52.8_ resulted in mass losses of 49.38% and 38.03% over a wide temperature range of 187.51–484.08 °C and 178.57–480.20 °C, respectively. This was potentially due to the added decomposition of the SCx hydrogel network. Increases in the concentration of Cr^3+^ decreased the final mass loss. Therefore, the mass loss rate of SCx exhibited an inverse relationship with the concentration of Cr^3+^ ions. This indicated that the formation of hydrogels with metals reduced the skeletal degradation of succinoglycan. The results also indicated that the metallohydrogel exhibited better thermal stability than pure succinoglycan.

#### 3.4.2. DSC Analysis

The DSC curves of succinoglycan and SCx are shown in Figure 6b, and the heat flow values corresponding to the endothermic peak temperature are listed in Table 2. The endothermic peak of DSC is typically due to the breakdown of hydrogen bonds and loss of hydroxyl groups when the sample is heated [55]. In the process of heating, the loss of water attached to the hydroxyl or carboxyl groups of succinoglycan leads to the breakdown of hydrogen bonds inside the hydrogels, which induces the phase transition of SCx. The melting temperature (T_m_) generally refers to the transition of the material from a crystalline state to an amorphous state [56], and thus, the transition temperature corresponds to the melting temperature T_m_. As shown in Figure 6b, the melting temperature (T_m_) of succinoglycan is observed as 92.17 °C. However, SCx exhibited a change to the maximum T_m_ corresponding to 117.99 °C based on the concentration of Cr^3+^. Furthermore, the endothermic enthalpy change (ΔH) increased in SCx when compared to that in succinoglycan. The viscosity of succinoglycan decreased sharply at temperatures exceeding 60 °C (Appendix A). The phenomenon was attributed to the order–disorder conformational transition wherein the chaotic form predominates and the viscosity decreases above 60 °C [57]. The decrease in the molecular weight induced by the irreversible change in viscosity and chain breakage or aggregate breakage can also led to a difference. The negatively charged carboxyl groups of succinoglycan were co-ordinated with Cr^3+^, and thus a higher change in enthalpy was required in SCx due to the hydrogel network induced by the addition of Cr^3+^ [53].

### 3.5. Microstructural Morphology of Metallohydrogels

FESEM images of the fractured surfaces of the hydrogels are shown in Figure 7. The pore size tended to decrease with increases in the concentration of Cr^3+^, thereby indicating that increases in the amount of Cr^3+^ led to a denser network inside SCx. The result indicated that the concentration of Cr^3+^ plays an important role in co-ordination with succinoglycan. The average pore size of each sample was defined using Image J software (version 1.49v NIH, USA). The pore sizes of SC_6.6_, SC_13.2_, SC_26.4_, and SC_52.8_ corresponded to 215.5 ± 29 μm, 138.5 ± 20.5 μm, 125 ± 32.5 μm, and 203.5 ± 16.3 μm, respectively. The pore diameter gradually decreased with increases in the concentration of Cr^3+^ and the smallest pore size was observed in SC_26.4_. Conversely, the pore size of SC_52.8_ exceeded that of SC_26.4_ despite the highest Cr^3+^ concentration. This was potentially due to the electrostatic repulsion inside the network due to an excessive number of Cr^3+^ cations. Thus, the pore size of the obtained SCx can be adjusted by changing the ratio of succinoglycan to (Cr^3+^).

### 3.6. Mechanical Properties of Metallohydrogels

#### 3.6.1. Oscillation Angular Frequency Sweep Test

The viscoelasticity of the hydrogels was evaluated via oscillation angular frequency sweep tests (Figure 8a). The experiment was swept from 0.1 to 100 rad/s at a fixed strain of 0.5%. The storage modulus (G′) corresponds to the value of the resistance to elastic deformation and represents the degree of structuring of the sample. Loss modulus (G″) is a measure of resistance to liquid flow, and it represents the viscosity properties of the test sample [58]. As shown in Figure 8a, all metallohydrogels exhibit storage modulus G′ values that exceed those of the loss modulus G″ at a given frequency. This implies that the hydrogel was in the state of an elastic solid gel state and not in a fluid sol state [59]. Increases in the number of crosslinks increase the storage modulus (G′) of the hydrogel, thereby indicating that the mechanical properties of the hydrogel depend on the number of crosslinks inside the hydrogel. Increases in the concentration of Cr^3+^ led to a gradual proportional increase in the value of G′. Conversely, it reached the maximum value in SC_26.4_ and subsequently decreased in SC_52.8_. Hence, the appropriate amount of Cr^3+^ exhibited a denser and harder hydrogel network due to metal co-ordination with several succinoglycans. Additionally, the results confirmed that the mechanical properties of the resulting metallohydrogel can be controlled by adjusting the concentration of the crosslinking density of SCx.

#### 3.6.2. Stress Oscillation Strain Amplitude Sweep Test

Stress oscillation strain amplitude sweep tests were conducted after fixing the frequency to 1.0 Hz to determine the limit point of the linear viscoelastic region. As shown in Figure 8b, the storage modulus (G′) of SCx stays constant in the strain range of 0.1–10%. Thus, this indicates that SCx can withstand high values of shear strain. Furthermore, this implies that SCx retained its gelling properties even at strains as high as 10%. When the strain exceeded 10%, the G′ value significantly decreased and the G″ value rapidly increased to generate a crossover point with values corresponding to 36% (SC_6.6_), 31% (SC_13.2_), 25% (SC_26.4_), and 22% (SC_52.8_). The network of each SCx was completely broken and converted into a sol state at that strain. Hence, the structure of the hydrogel network was completely damaged and exhibited fluid-like properties [60]. The difference in the critical strain values of the four hydrogels was potentially due to the flexibility of the gel network. In particular, SC_52.8_, with intersections at the lowest strain, indicate hard gels, and SC_6.6_, with intersections at higher strains, indicate soft gels. Thus, the flexibility of the formed SCx can be controlled by changing the concentration of Cr^3+^.

#### 3.6.3. Mechanical Properties of SCx in Terms of Compression Strength

Compressive tests were conducted to obtain compressive strain curves for investigating the effect of different Cr^3+^ concentrations on the mechanical properties of SCx (Figure 9). The samples were compared at a strain of 40%. The order of compressive strength for the SCx hydrogel corresponded to SC_26.4_: 0.1945 MPa > SC_13.2_: 0.749 MPa > SC_52.8_: 0.0258 MPa > SC_6.6_: 0.0190. For SCx, increases in the concentration of Cr^3+^ at 40% strain resulted in a proportional improvement in compressive strength by approximately 10 times that in the range of 0.0190 MPa (S_C6.6_) to 0.1945 (SC_26.4_). Conversely, SC_52.8_ exhibited a significant decrease in the mechanical strength corresponding to 0.0258 MPa, thereby indicating that electrostatic repulsions can occur inside the network due to excessive concentrations of Cr^3+^. This indicated that the strongest hydrogel formation occurred at an optimal concentration (26.4 mM) of Cr^3+^ as opposed to the highest concentration. In conclusion, the crosslinking effect of Cr^3+^ was the highest in SC_26.4_, and the results confirmed that the mechanical strength can be effectively improved or controlled by controlling the concentration of Cr^3+^ when preparing a hydrogel with metal.

### 3.7. Effect of pH of Metallohydrogel on Gelation

The effect of the manufacturing pH on the rheological properties of SCx was examined. As shown in Figure 10a, the range of gel formation was determined by the pH. The experiment was compared by considering six points in the range of pH 1 to 11. The results indicated that G′ exceeded G″ in the weak acidic pH range of 3 to 5, thereby indicating that gelation occurred in the pH range of 3–5. However, the sample lost its elastic properties outside the aforementioned range of pH. This is demonstrated by the appearance of the intersection of G′ and G″ by changing the rheological behavior (Appendix A). The loss tangent (tanδ (G″/G′)) is a dimensionless parameter that measures the ratio of energy lost to energy stored in a cyclic deformation [53]. As shown in Figure 10b, strongest hydrogels are formed with the smallest tanδ values at pH 5, and this confirms that pH is an important factor that affects the metal co-ordination of succinoglycan and Cr^3+^. This is related to the nature of interactions based on the structure of the chromium species formed in water and formation of carboxyl groups. In extremely acidic conditions (such as a pH less than 3), Cr^3+^ predominantly exists in the monomeric form, and (Cr(H_2_O)_6_)^3+^ is co-ordinated with six water molecules [61,62]. Given that the Cr^3+^ monomer is small, it cannot be effectively co-ordinated with the carboxyl group (pK_a_~3.8) of succinoglycan, where protonation of the carboxylic group can also impede their interactions with Cr^3+^ [63]. However, a higher pH range from 3 to 5 can lead to olate forms of Cr^3+^ (dimeric, trimeric, and tetrameric oligomers), which can lead to successful co-ordination with succinoglycan [43]. In neutral or alkaline conditions with higher pH at which gel formation occurs, chromium ions changed to green in aqueous solution and were converted to water-insoluble hydroxide Cr(OH)_3_ [64]. In summary, only in the pH range of 3–5 (in which the gel was formed) did Cr^3+^ in the form of olates form an effective co-ordination with the deprotonated carboxyl group of succinoglycan.

At pH 1, effective crosslinking with succinoglycan did not occur although gel formation was observed as soon as the pH of the solution was changed to 5, and the same phenomenon occurred when the pH of the solution prepared at pH 9 was changed to 5 (Appendix A). The phenomenon can be attributed to the aforementioned pH-dependent behavior of Cr^3+^.

In both cases, the solution state was changed to a gel state via changes in pH, and rheological properties and mechanical strength were significantly increased (Figure 11a,b). However, the mechanical strength was not completely recovered when compared to that of the hydrogel initially prepared at pH 5 (Appendix A). This was potentially because Cr^3+^ (which corresponded to a monomeric form at a low pH and an insoluble form at a high pH) did not change sufficiently in a short period of time even with a pH change. However, the sample in which pH changed from 5 to a strong acid (pH 1) or strong base (pH 9) remained a gel (Figure 11c,d). A slight increase was observed in the final pH 1 gel because carboxyl groups that were not involved in the crosslinking with Cr^3+^ became uncharged, thereby decreasing counter ion pressure and the swelling degree of the gel [43]. The results indicate that Cr^3+^ and succinoglycan are strongly crosslinked, and thus, the already crosslinked metallohydrogel is less affected by pH.

### 3.8. Swelling Behavior of Metallohydrogel

The swelling ability of a hydrogel is an important parameter in evaluating its properties. The swelling ratio of SCx was examined in distilled water at 25 °C and plotted with respect to time. As shown in Appendix A, the swelling ratio of SCx increases rapidly and then reaches an equilibrium after 5 min. Further, the swelling ratio decreased with increases in the concentrations of Cr^3+^. The mechanism underlying the phenomenon was closely related to crosslink density. The mechanism underlying the phenomenon was closely related to high crosslinking density due to the excessive concentration of Cr^3+^, which inhibits the hydrophilicity of succinoglycan by co-ordinating with the O-H bonds of succinoglycan, thereby suppressing the swelling [34]. Additionally, a UV–Vis spectrophotometer was used to confirm whether chromium ions were released after immersion in distilled water (Appendix A). Spectra were observed from 200 to 700 nm at 2 h intervals over 48 h. Peaks did not appear in the solution immersed in SCx, as shown in Appendix A, where the intrinsic peaks of the Cr^3+^ (5 mM) solution appeared at 570 nm and 415 nm. This confirmed that Cr^3+^ did not release Cr^3+^ when combined with succinoglycan. The results confirmed that Cr^3+^ in the hydrogel was not released even if swelling proceeded.

## 4. Conclusions

In this study, succinoglycan-based metallohydrogels (SCx) were successfully obtained via the metal co-ordination of Cr^3+^ with succinoglycan. Among other metal ions, Cr^3+^ effectively formed a co-ordinated metallohydrogel with succinoglycan at lower concentrations. The characteristics of SCx were investigated via FTIR, XRD, TGA, DSC, FESEM, and rheological measurements. In particular, FTIR spectroscopy confirmed that the carboxyl and hydroxyl groups of succinoglycan played an important role in metal co-ordination. The change in the thermal stability of succinoglycan was observed via performing TGA and DSC. The results were confirmed by decreases in final weight loss with increases in the concentration of Cr^3+^ (6.6–52.8 mM). Rheological measurements and FESEM indicated that pore size and mechanical properties can be controlled by adjusting the concentration of Cr^3+^. Additionally, succinoglycan metallohydrogellation occurred in a pH-dependent manner. Gelation can also be adjusted based on the initial pH of the hydrogel formulation. A gel was obtained when the pH at preparation ranged from 3 to 5. Furthermore, when the pH of the solution was in the range of 1–9, the pH changed to 5 and gelation occurred. However, crosslinking of the gel was not affected when the gel was stably formed at pH 5 even if the pH was subsequently changed to strongly acidic or basic. This indicated that when crosslinking of the hydrogel with metal is formed, it is stable even under extreme pH conditions. This was attributed to the pH-dependent behavior of Cr^3+^ in aqueous solutions and to the carboxyl groups of succinoglycan that affect metal co-ordination.

In conclusion, the pH-dependent metallohydrogel could increase the thermal stabil-ity of succinoglycan, control pore size as well as mechanical strength by adjusting the concentration of Cr^3+^. It could be used as a new soft matrix for bioseparation by tuning the Cr^3+^ concentration to control pore size and mechanical strength. In addition, the proposed pH-dependent metallohydrogel offers a promising method for the controllable gelation of succinoglycan with a metal ion and might be applied to monitor Cr^3+^ in drinking water or living cells.

## Figures and Tables

**Figure 1 polymers-13-00202-f001:**
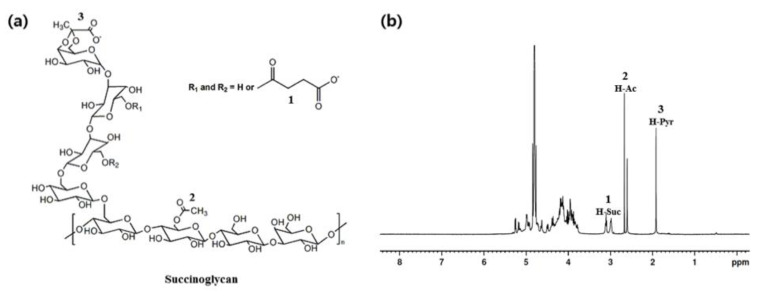
Structure of (**a**) succinoglycan and (**b**) ^1^H-NMR spectrum of succinoglycan.

**Figure 2 polymers-13-00202-f002:**
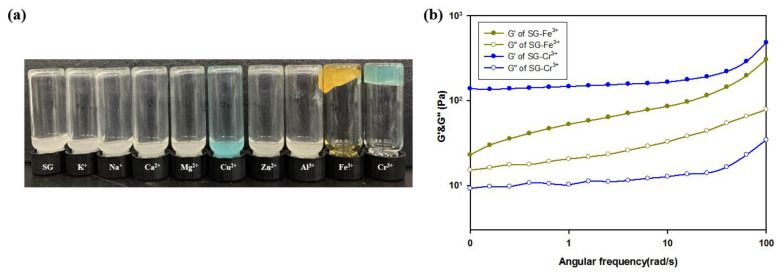
Gelling performance of succinoglycan (SG) with respect to the addition of various metal ions: (**a**) Inverted vial test after mixing various metal ion solutions (0.25 M) with aqueous succinoglycan solution (wt. 2%), and (**b**) Frequency sweep of aqueous SG solution containing metal ions (SG-Fe^3+^ and SG-Cr^3+^).

**Figure 3 polymers-13-00202-f003:**
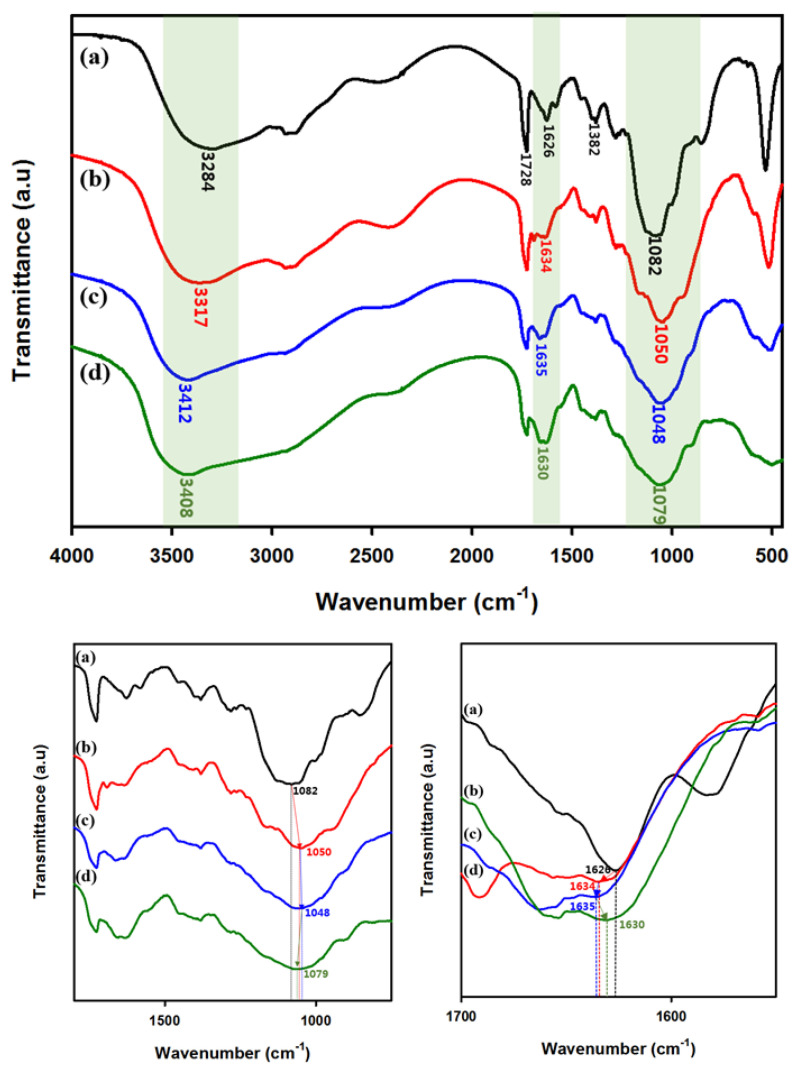
Fourier transform infrared (FTIR) spectra: (**a**) succinoglycan, (**b**) SC_13.2_, (**c**) SC_26.4_, and (**d**) SC_52.8_.

**Figure 4 polymers-13-00202-f004:**
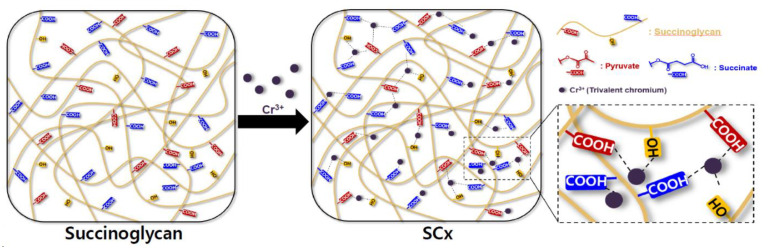
Schematic of a hydrogel crosslinked by complexation of succinoglycan with Cr^3+^ in aqueous solution.

**Figure 5 polymers-13-00202-f005:**
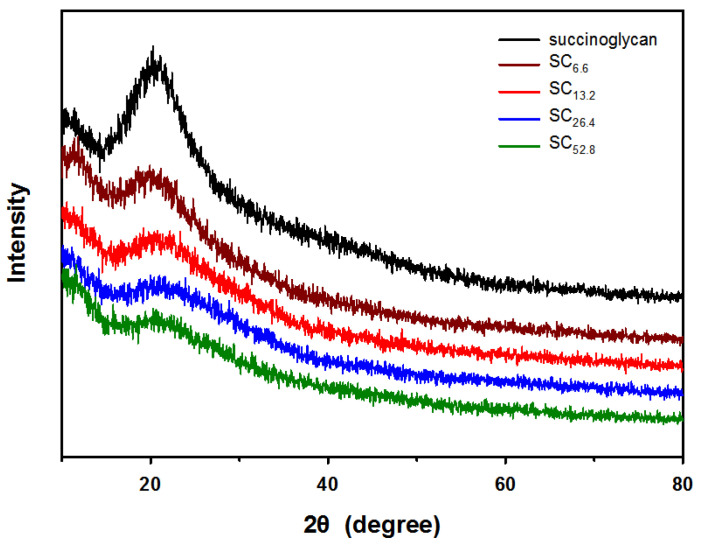
Changes in X-ray diffraction (XRD) patterns of succinoglycan due to hydrogel formation by addition of Cr^3+^.

**Figure 6 polymers-13-00202-f006:**
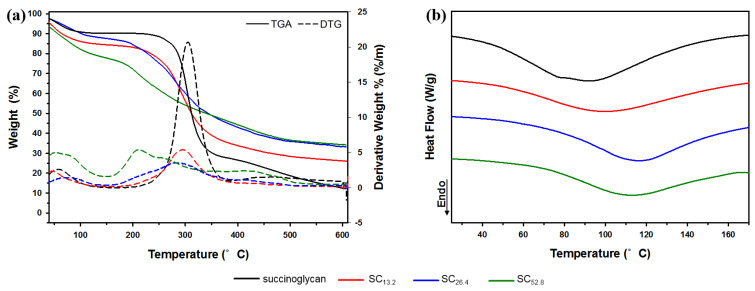
Comparison of the thermal analysis for SCx with respect to different Cr^3+^ concentrations: (**a**) thermogravimetric analysis (TGA) and (**b**) differential scanning calorimetry (DSC).

**Figure 7 polymers-13-00202-f007:**
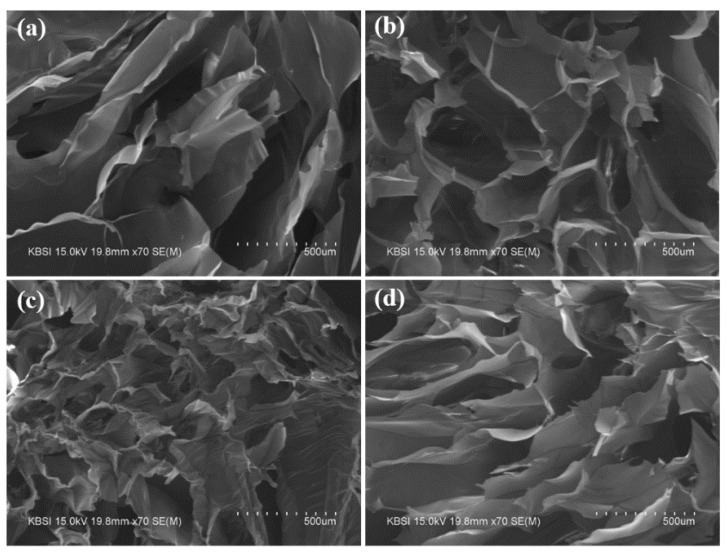
Field emission scanning electron microscopy (FESEM) micrographs of the fractured cross sections of (**a**) SC_6.6_, (**b**) SC_13.2_, (**c**) SC_26.4_, and (**d**) SC_52.8_.

**Figure 8 polymers-13-00202-f008:**
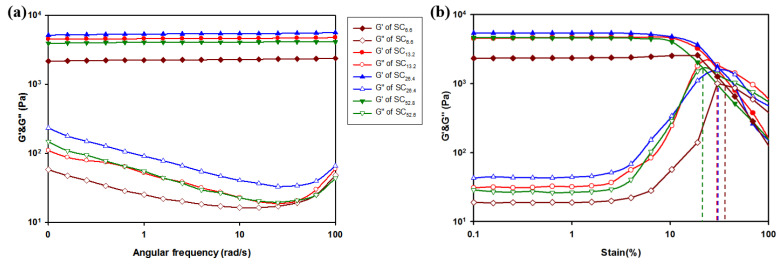
Rheological measurement for SCx: (**a**) oscillation angular frequency sweep test and (**b**) stress oscillation strain amplitude sweep test. Storage modulus (G′, filled symbols) and loss modulus (G″, empty symbols) of hydrogel was measured as a function of angular frequency (0.1–100 rad/s) and strain sweep (0.1–100%).

**Figure 9 polymers-13-00202-f009:**
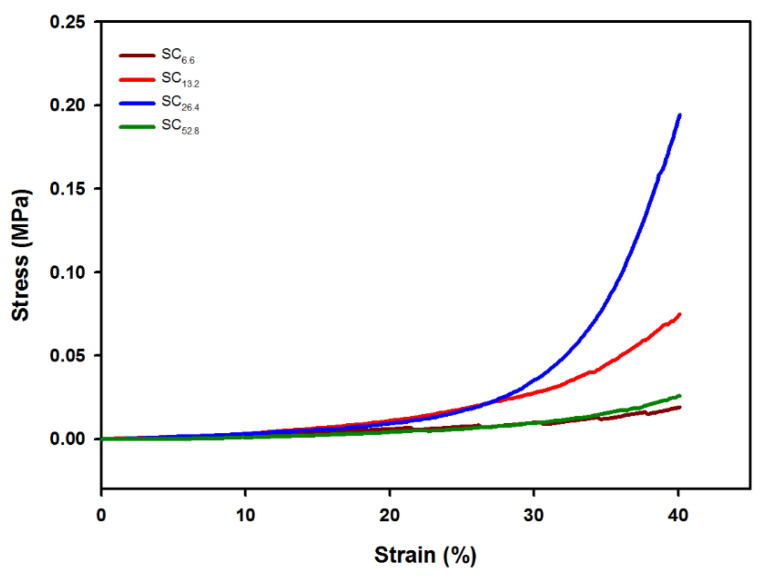
Compressive strain curves of SCx with different Cr^3+^ concentrations.

**Figure 10 polymers-13-00202-f010:**
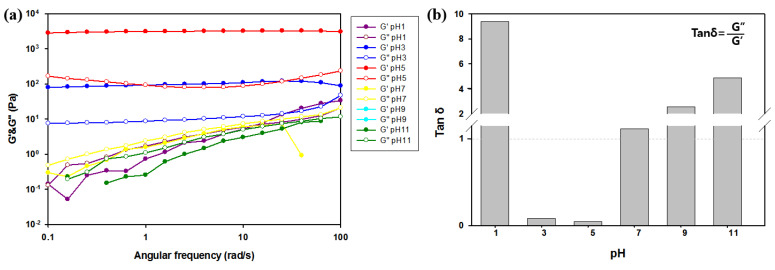
pH-Dependent gelation effect and mechanical properties of SC_26.4_. (**a**) Storage modulus (G′, filled symbols) and loss modulus (G″, empty symbols) of hydrogels. (**b**) Loss tangent (tan δ) of hydrogels.

**Figure 11 polymers-13-00202-f011:**
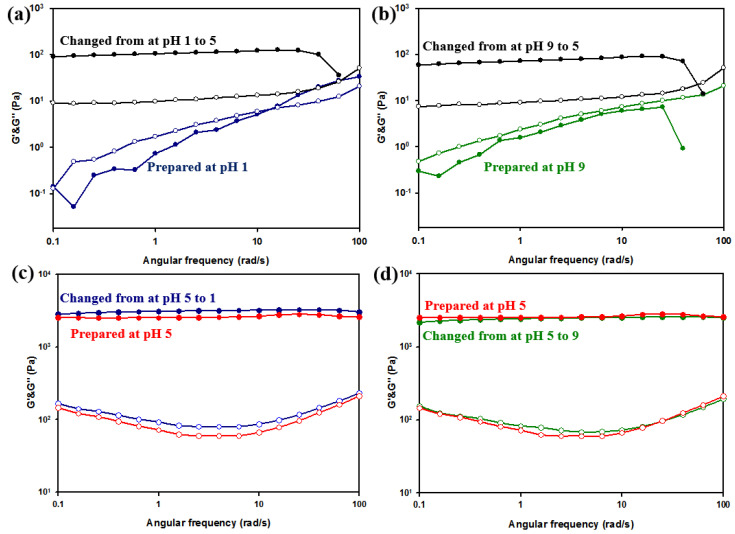
Gelation effect and mechanical properties of SC_26.4_ based on pH changes: (**a**) from pH 1 to 5, (**b**) from pH 9 to 5, (**c**) from pH 5 to 1, and (**d**) from pH 5 to 9. Storage modulus (G′, filled symbols) and loss modulus (G′′, empty symbols) of hydrogels were measured as a function of angular frequency (0.1–100 rad/s).

**Table 1 polymers-13-00202-t001:** Composition of SCx.

Sample Name	Succionglycan (wt %)	Cr^3+^ (mM)
SC_6.6_	1	6.6
SC_13.2_	1	13.2
SC_26.4_	1	26.4
SC_52.8_	1	52.8

**Table 2 polymers-13-00202-t002:** Thermal properties of metallohydrogels.

Sample Name	TGA				DSC	
	First Mass Loss Stage	Second Mass Loss Stage			Endothermic Peak	
	Mass Loss (%)	Onset Temperature (%)	DTG (°C)	Mass Loss (%)	Temperature (°C)	Heat Flow (W/g)
succinoglycan	8.05	246.14	304.98	60.64	92.17	−1.517
SC_13.2_	12.93	213.12	295.68	53.08	101.40	−1.837
SC_26.4_	10.47	187.51	285.25	49.38	117.99	−1.829
SC_52.8_	19.15	178.57	210.95	38.03	114.13	−1.998

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
