# Peer review of "Fabrication and Characterization of Polysaccharide Metallohydrogel Obtained from Succinoglycan and Trivalent Chromium"

_polymers, 2021, doi:10.3390/polym13020202_

Round 1

Reviewer 1 Report

Authors describe the formation of succionaglucan/Cr3+ hydrogel. Oberall, the study could be interesting but there are several part that must be improve and some lacks that must be completed before being considered for its publication in Polymers.

The abstract must clear describe the interest and the advance from the state of art of these materials.

Cr+3 is less toxic than Cr+3, ok, but it is toxic so please clarify how it could be interesting for biomedical applications, or which other applications are expected for this material?

Figure 6. indicate the direction of endo or exo at the heat flow

Show the complete thermogram for DSC . Considering the shape, are the peaks on FIG 6 for DSCwater?

Lines311 “The pore sizes of SC6.6, SC13.2, SC26.4, and SC52.8 corresponded to 215.5±29, 138.5±20.5, 125±32.5, and 203.5±16.3, respectively.” Units??

How many repetitions of the mechanical test have been carried out?

Author Response

Response to Reviewers’ Comments

Reviewer 1’s  comments :

Authors describe the formation of succionaglucan/Cr3+ hydrogel. Oberall, the study could be interesting but there are several part that must be improve and some lacks that must be completed before being considered for its publication in Polymers.

(1) The abstract must clear describe the interest and the advance from the state of art of these materials. Cr+3 is less toxic than Cr+3, ok, but it is toxic so please clarify how it could be interesting for biomedical applications, or which other applications are expected for this material?

☞ As I had mentioned before, Cr3+ is less toxic than Cr6+ (Please see the references 37-39). And we showed that no chromium ions (Cr3+) were released after immersing the SCx in distilled water via the UV-Vis analysis as shown in Figure S6 where the intrinsic peaks of the Cr3+ (5 mM) solution appeared at 570nm and 415nm. We had also suggested several studies on its (Cr3+) applications for bioimaging to living cells [40-42] were reported. So, basically Cr3+ would have potential for biomedical applications even though it is less toxic. So, I think, as future possible application of this pH-dependent metallohydrogel, this gel might offer an interesting method for rapid removing (or sensing) trivalent chromium (Cr3+) using succinoglycan solution for environmental or biomedical purpose, since the swelling of SCx could rapidly occur and reach equilibrium soon after 5 minutes depending on the pH-condition. As SCx exhibited strong gelation in the pH 5-7 range, it would also be used to monitor Cr3+ in drinking water or living cells. So, succinoglycan based metallohydrogel would have potential to be used as a soft sensor for Cr3+ detection in biological systems. Furthermore, since we also observed that succinoglycan-based metallohydrogel had different pore size as well as mechanical strength by adjusting the concentration of Cr3+, it could be used as a soft matrix for bioseparation.

We added some explanations in the abstract according to your suggestion.

However, this manuscript mainly focused on the novel fabrication and characterization of metallohydrogel using succinoglycan and trivalent chromium.

[37] GREVATT, Peter. http://www. epa. gov/IRIS/toxreviews/0028-tr. pdf, 1998.

[38] WILBUR, Sharon, et al. Toxicological profile for chromium. 2012.

[39] EPA, U. S. Integrated Risk Information System, National Center for Environmental Assessment, Washington, DC, USA, 1999.

[40] ERDEMIR, Serkan; KOCYIGIT, Ozcan. Talanta, 2016, 158: 63-69.

[41] JIANG, Tiantian, et al. Spectrochimica Acta Part A: Molecular and Biomolecular Spectroscopy, 2020, 245: 118903.

[42] DIAO, Quanping, et al. Spectrochimica Acta Part A: Molecular and Biomolecular Spectroscopy, 2016, 156: 15-21.

After>

Abstract

In the present study, a polysaccharide metallohydrogel was successfully fabricated using succinoglycan and trivalent chromium and was verified via Fourier transform infrared spectroscopy, differential scanning calorimetry analysis, thermogravimetric analysis (TGA), field emission scanning electron microscopy, and rheological measurements. Thermal behavior analysis via TGA indicated that the final mass loss of pure succinoglycan was 87.8% although it was reduced to 65.8% by forming a hydrogel with trivalent chromium cations. Moreover, succinoglycan-based metallohydrogels exhibited improved mechanical properties based on the added concentration of Cr3+ and displayed a 10 times higher compressive stress and enhanced storage modulus (G’) of 230% at the same strain. In addition, the pore size of the obtained SCx could be adjusted by changing the concentration of Cr3+. Gelation can also be adjusted based on the initial pH of the metallohydrogel formulation. This was attributed to crosslinking between chromium trivalent ions and hydroxyl/carboxyl groups of succinoglycan, each of which exhibits a specific pH-dependent behavior in aqueous solutions. It could be used as a soft sensor to detect Cr3+ in certain biological systems, or as a soft matrix for bioseparation that allows control of pore size and mechanical strength by tuning the Cr3+ concentration.

(3) Figure 6. indicate the direction of endo or exo at the heat flow

☞ As your comments, we indicated the direction of endo in Figure 6 of the revised manuscript. Thanks.

(4) Show the complete thermogram for DSC . Considering the shape, are the peaks on FIG 6 for DSCwater?

☞ Figure 6 actually showed the complete thermogram for DSC. Normally, DSC endothermic peaks of succinoglycan were measured within 20-180 °C (International Journal of Biological Macromolecules, 2019, 134: 1013-1021, Food Hydrocolloids, 2015, 45: 18-29). In the case of other polysaccharides, DSC study was performed in a similar temperature range (Polymer Crystallization, 2019, 2.6: e10092). The peak shapes are normal in case of polysaccharide-based hydrogel. The peaks on Figure 6 are real peaks that could be influenced by water. As you knew, DSC was actually measured using a dried sample. As shown in Table 2, succinoglycan and SCx had endothermic peaks at 92.17°C and 101.4-117°C, respectively. Those endothermic peaks of DSC are typically due to the breakdown of hydrogen bonds and loss of hydroxyl groups when the sample is heated (Food Hydrocolloids, 2015, 45: 18-29). In the process of heating, the loss of water attached to the hydroxyl or carboxyl groups of succinoglycan leads to the breakdown of hydrogen bonds inside the hydrogels, which induces the phase transition of SCx. Therefore, though the broad endothermic peaks can be influenced by water, all are real peaks representing the different network structures induced by adding trivalent chromium.

(5) Lines311 “The pore sizes of SC6.6, SC13.2, SC26.4, and SC52.8 corresponded to 215.5±29, 138.5±20.5, 125±32.5, and 203.5±16.3, respectively.” Units??

☞ Unit of pore sizes was μm and we corrected it in the revised manuscript. Thanks for your indication.

(6) How many repetitions of the mechanical test have been carried out?

☞ We performed each measurement in triplicate. We added this in the revised manuscript. Thanks.

Thank you very much for your critical and useful suggestions.

Reviewer 2 Report

The authors improved the manuscript according to my suggestions. Thus, the manuscript can be accepted for publication.

Author Response

Thank you very much for your kind suggestions.

This manuscript is a resubmission of an earlier submission. The following is a list of the peer review reports and author responses from that submission.

Round 1

Reviewer 1 Report

Authors describe the formation of succionaglucan/Cr3+ hydrogel. The study is interesting but there are several part that must be improve and some lacks that must be completed before being considered for its publication in Polymers.

First of all, a sweeling study of this hydrogels must be added, in addition to an analysis of the Cr3+ during the process, is the Cr3+ released during the sweeling?? This is very important since the cation could be toxic.

Abstract and Intorduction. The interest of this work must be clear. Why is this study interesting? The improvement proposed with this work must be describe.

Line 24. “many potentials” this does not give more informations, the potential must be referred to something.

Line 25. “it is extensively ….” The previous sentence is in plural, so in the second one to which is the polysaccharide that you are describing?

Ln 26-28. References are needed

Ln 36-39. References are needed.

The use of Cr3+ must be justify, this is a toxic cation, so the potential uses of this hydrogel are going to be limited, which are the possible uses of this system.

The summary at the end of the introduction part miss many of the reported experiments, so please improve the description.

Materials. Provide the purity of the materials used.

2.3. This section has to be removed, they are not a synthesis. In this section a normal solutions are described so either you justify the description because of some relevant phenomenon occurs in it, or you have to removed, readers know how to make a solution.

It is mandatory to provide the molecular weight of the polysaccharides used in this study.

The preliminary study of the cations must be described in the experimental section

The discussion on XRD studies has to be improved, compare the result with the existing bibliography.

DSC study is not a thermal stability study please remove stability from the title

Lin 182-184- I believe that Polymers readers know what TGA and DTG is, so remove this part.

The temperatures in which each step start and end must be added to the discussion.

Considering the broad bands presented in the DSC, and the TGA data, it seems that what are you considering as Tm is water and not a proper Tm, please provide a first and second runs, in order to confirm that the Tm is real.

Mechanial properties. The jump of the SC 52.8 has to be explained.

Reviewer 2 Report

The current manuscript of Jung and coworkers describes the synthesis of hydrogels based on metallopolymers. For this purpose, chromium was added to a polysaccharide in order to obtain hydrogels, which feature also pH-dependent behavior. The research is performed in a suitable manner, however, I have same major points, which needs to be addressed.

1.) The are several small stylistic errors. Please revise the manuscript accordingly.

2.) What is the resolution of the IR-spectrometer? Is the observed change really measureable by your instruments?

3.) Can you compare these changes with small molecular compounds, e.g., metal complexes?

4.) What is information obtained from the XRD? I do not see any benefit of this measurement for the current study.

5.) I have a significant problems with the thermal properties and their interpretation. The DSC signal fits perfectly to the first loss in the TGA. Thus, the observed endothermic signal would fit to the degradation and not to a Tm of the polymer. Can you prove your statement?

6.) Furthermore in this context, the first loss in the TGA would fit to an evaporation of water (not complexed one), residual water of the hydrogels seems to be a better explanation.

7.) In this context, what is the amount of water within the hydrogels and did you measure water uptake for the dried samples in order to investigate the suitable amount of water?